# Scoping Review of Machine Learning and Patient-Reported Outcomes in Spine Surgery

**DOI:** 10.3390/bioengineering12020125

**Published:** 2025-01-29

**Authors:** Christian Quinones, Deepak Kumbhare, Bharat Guthikonda, Stanley Hoang

**Affiliations:** Department of Neurosurgery, Louisiana State University Health Shreveport, Shreveport, LA 71103, USA

**Keywords:** artificial intelligence, machine learning, patient-reported outcomes, spine surgery, outcome measures, literature review, health informatics

## Abstract

Machine learning is an evolving branch of artificial intelligence that is being applied in neurosurgical research. In spine surgery, machine learning has been used for radiographic characterization of cranial and spinal pathology and in predicting postoperative outcomes such as complications, functional recovery, and pain relief. A relevant application is the investigation of patient-reported outcome measures (PROMs) after spine surgery. Although a multitude of PROMs have been described and validated, there is currently no consensus regarding which questionnaires should be utilized. Additionally, studies have reported varying degrees of accuracy in predicting patient outcomes based on questionnaire responses. PROMs currently lack standardization, which renders them difficult to compare across studies. The purpose of this manuscript is to identify applications of machine learning to predict PROMs after spine surgery.

## 1. Introduction

Research in spine surgery has been impacted by the recent rise in artificial intelligence (AI). Machine learning (ML) is a subset of AI that functions to predict outputs based on given inputs. In medical research, input data may include any combination of the following: patient demographics, spinal pathology, imaging characteristics, surgical characteristics, comorbidities, and patient-reported outcome measures (PROMs) [1]. Examples of outputs are complications, functional outcomes, surgical success, hospitalization characteristics, readmission rates, reoperation rates, survival prediction, cost prediction, and rehabilitation needs. One outcome in which ML is particularly applicable is in predicting PROMs after spine surgery.

When being evaluated for spine surgery, an important consideration is the degree of improvement that a patient experiences after surgical intervention. This question can be answered by comparing preoperative and postoperative PROMs. The original PROMs developed for use in spine surgery are currently referred to as “legacy outcome measures” and include the Oswestry Disability Index (ODI), Neck Disability Index (NDI) [2], Visual Analog Scale (VAS), Short Form Health Survey (SF-36 or SF-12), Japanese Orthopaedic Association (JOA) score, Roland-Morris Disability Questionnaire (RMDQ), EuroQol-5D (EQ-5D), and Scoliosis Research Society (SRS) questionnaire [3]. These surveys provided the foundation for defining patient-oriented, clinically significant outcomes that assess quality of life after spine surgery [4]. To quantify a standard for expected PROM improvements, clinicians defined the Minimal Clinically Important Difference (MCID) for these PROMs [5]. Due to variations in spinal pathology, surgical interventions, patient demographics, and the intrinsic disadvantages of PROMs such as time to completion, there has been a lack of consensus on which PROMs to utilize. A 2022 literature review reported the presence of 206 unique spine-specific PROMs [6]. To address this, the National Institute of Health developed the Patient-Reported Outcomes Measurement Information System (PROMIS) in an attempt to standardize PROMs and simplify their administration [7].

The decision to proceed with spine surgery is often complex, largely because there are no definitive guidelines or universal indications for when surgery is appropriate. The use of ML to accurately predict patient outcomes grants surgeons another tool to more confidently advise patients on surgical outcomes [8]. The purpose of this manuscript is to describe the extent to which ML has been used to predict PROMs after spine surgery.

## 2. Materials and Methods

A scoping review of the literature per the Preferred Reporting Items for Systematic Reviews and Meta-Analyses for Scoping Reviews (PRISMA-ScR) guidelines [9] was carried out in Web of Science, PubMed, and EMBASE on October 8, 2024. A combination of MeSH terms and keywords related to patient-reported outcomes and spine surgery were used. The search criteria for PubMed were as follows: (“Machine Learning” [MeSH] OR “Artificial Intelligence”) AND (“Patient Reported Outcome Measures” [MeSH] OR “Patient-reported outcomes”). The search criteria for Web of Science were as follows: (“Machine Learning” OR “Artificial Intelligence”) AND (“Patient Reported Outcome Measures” OR “Patient-reported outcomes” OR “PROMs” OR “Quality of Life” OR “Health Outcomes”) AND (“Spine” OR “Spinal Surgery”). The search criteria for EMBASE were as follows: (‘machine learning’/exp OR ‘machine learning’ OR ‘artificial intelligence’/exp OR ‘artificial intelligence’) AND (‘patient reported outcome’/exp OR ‘patient reported outcome’ OR ‘quality of life’/exp OR ‘quality of life’ OR ‘patient-reported outcomes’ OR ‘proms’ OR ‘health outcomes’/exp OR ‘health outcomes’) AND (‘spine’/exp OR ‘spine’ OR ‘spinal surgery’/exp OR ‘spinal surgery’ OR ‘spine surgery’/exp OR ‘spine surgery’) AND [english]/lim.

English articles published from 1994 to 2024 were selected. One researcher (C.Q.) assessed the manuscripts for eligibility under the supervision of another researcher (D.K.). In cases of disagreements or uncertainties requiring further clarification, the senior author (S.H.) was consulted and a consensus was reached during research team discussions. The inclusion criteria consisted of studies that utilized ML tools to predict postoperative PROMs for patients who underwent spine surgery. Studies that did not employ ML to predict PROMs were excluded. Data extracted included the ML method used, spine pathology, number of patients, features used for model prediction, and ML performance.

## 3. Results

### 3.1. Search Results

The initial search yielded 648 articles; 60 repeats were removed, resulting in 588 unique articles for screening. Of the 37 articles that met the initial screening criteria, twelve non-surgical studies were excluded. The remaining 25 articles were further assessed for eligibility, with 3 excluded for not predicting postoperative PROMs [10,11,12]. A total of 22 articles were included in the qualitative synthesis (Figure 1).

### 3.2. Study Details

Seven articles predicted outcomes in cervical spine pathologies [13,14,15,16,17,18,19]. Three articles predicted outcomes for thoracolumbar pathologies [20,21,22]. Eleven articles predicted outcomes for lumbar spine pathology [23,24,25,26,27,28,29,30,31,32,33]. One study predicted outcomes for all levels of spinal pathology [34]. The postoperative timeline for PROM prediction ranged from 6 weeks to 24 months (Table 1).

A total of twenty-one PROMs were reported. Seven articles reported the ODI [19,20,23,24,25,26,31], and four articles reported the VAS [13,23,25,29], mJOA [14,15,18,19], and numeric rating scale (NRS) [20,24,30,31]. Three reported the JOA [13,25,29], NDI [13,15,17], core outcome measure index (COMI) [26,27,34], and SF-36 [15,16,26]; two articles reported the EQ-5D [13,17] and SRS [21,22]; and one article reported the EuroQol [25], Physical Component Summary (PCS) [26], PROMIS-PF [28], SF-6D [14], Mental Component Summary (MCS) [15], Mental Disability Index (MDI) [15], Disabilities of the Arm, Shoulder, and Hand (DASH) [15], North American Spine Society (NASS) [15], Japanese Orthopedic Association Back Pain Evaluation Questionnaire (JOABPEQ) [29], neck pain [17], and pain symptoms specific to quality of life, social disability, and work disability [34]. Table 2 provides a categorical breakdown and brief description of PROMs.

The features used for model prediction were demographics in all but one study [33]. Surgical characteristics were used in ten studies [13,14,18,19,21,23,27,29,32,34]. Spinal pathology characteristics were used in ten studies [13,14,16,18,19,21,23,26,28,31]. American Society of Anesthesiologist (ASA) classification was used in six studies [13,20,23,24,28,31]. Physical exam findings were used in seven studies [13,14,15,16,17,18,19]. Past medical history (including surgical history) was used in five studies [19,26,30,31,34]. Preoperative opioid use was used in four studies [22,24,28,32]. Hospitalization details were used in three studies [22,24,27]. Social history (including employment details) was used in two studies [23,25]. One study used geographic details [28].

Sixty unique ML models were used in the relevant studies. The most frequently used model was support vector machine (SVM), which was used in eight studies [13,14,15,16,18,25,28,29]; logistic regression (LR), which was used in seven studies [13,14,16,21,23,27,29]; and RF, which was used in six studies [14,16,22,26,27,31]. Decision tree was used in four studies [13,14,21,25], and elastic net (EN) was used in three studies [22,28,32]. Least absolute shrinkage and selection operator (LASSO) regression was used in three studies [17,30,34], and neural network was used in three studies [14,28,31]. The remaining ML models included Bayesian generalized linear models (BGLMs), boosted LR, extra trees, extreme gradient boosted trees, regression tree, Tree—AS, boosting, chi-squared, deep learning, dimensionality reduction factor analysis, EN penalized LR, EN regularization, EN, generalized additive models, generalized boosted, generalized boosted machines, generalized linear mixed model, k-nearest neighbors, linear—AS, multilayer perceptron, multivariable adaptive regression splines, multivariate linear regression, partial least squares, principal component analysis, ridge regression, simple BGLMs, single-layer artificial neural networks, stepwise regression, and stochastic gradient boosting. Model performance was most frequently reported as Area Under the Curve (AUC), which was reported in sixteen studies [13,14,15,16,18,20,22,23,24,25,26,27,28,31,32,33]. Model performance was also reported as the mean absolute error (MAE) in three studies [21,29,34]. The remaining performance measures included mean bootstrapped R2 [30], MMA [19], and coefficients [17].

### 3.3. Key Results

Park et al. best predicted 3- and 24-month VAS after cervical spine decompression with LR with an AUC of 0.762 and 0.773, respectively [13]. Pedersen et al. used seven ML models to predict EQ-5D, ODI, VAS leg pain (LP), VAS back pain (BP), and return to work after lumbar spine surgery with a mean AUC of 0.82, 0.75, 0.73, 0.81, and 0.84, respectively [25]. Ve et al. employed a deep learning model to predict the ODI with an AUC of 0.84 and NRS BP improvement with an AUC of 0.9 [23]. Berjano et al. predicted postoperative ODI with a combination of preoperative ODI, SF-36 Physical Component Summary (PCS), and COMI Back with an AUC of 0.808 [26]. Halicka et al. used LR to predict an AUC of 0.63, 0.72, and 0.68 for COMI, BP, and LP, respectively [27]. Karhade et al. utilized LR, neural networks, and EN penalized LR to predict PROMIS physical function, pain interference, and pain intensity, achieving AUCs of 0.75, 0.71, and 0.71, respectively, with the EN penalized LR achieving an AUC of 0.69 [28]. Merali et al. used random forest random forest (RF) to predict SF-6D and mJOA with an of AUC of 0.85, 0.83, and 0.87 at 6, 12, and 24 months, respectively [14]. Rigoard et al. found that changes in the Modified Clinical Response Index were the most accurate indicator of Patient Global Impression of Change, with an AUC of 0.853 [33]. This was higher compared to the AUC for changes in the Hospital Anxiety and Depression Scale (HADS) (0.780), ODI score (0.737), Numerical Pain Rating Scale (NPRS) (0.704), EQ-5D index (0.698), and Pain Mapping Intensity score (0.672). Grob et al. used EN regularization to predict ODI, NRS BP, and LP with an AUC of 0.70, 0.72, and 0.70, respectively [20]. Zhang et al. used SVM to predict SF-36 PCS and Mental Component Summary (MCS) with an AUC of 86.4 and 89.8, respectively [15]. Gupta et al. used gradient boosting to predict an MAE of 0.47 and 0.55 for SRS-pain prediction and SRS self-image prediction, respectively [21]. Yagi et al. used an assemblage of the top five performing algorithms to predict JOABPEQ and VAS scores following lumbar spine surgery, with MAE values ranging from 9.3 to 16.5 [29]. Muller et al. used LASSO to predict COMI subdomains for back and neck pain with an MAE of 2.1 and 1.8, respectively [34]. Khan et al. used generalized boosted models and multivariable adaptive regression models to obtain predictions with an AUC of 0.77 and 0.78 for 24-month postoperative MCS and PCS, respectively [16]. Finkelstein et al. used LASSO regression to predict NRS after lumbar surgery with a mean bootstrapped R2 of 0.12 [30]. Liew et al. was the only study evaluating cervical radiculopathy [17]. This same study used four ML models to predict the NDI and EQ-5D. In this study, stepwise regression yielded the highest accuracy for the NDI, EQ-5D, and neck pain 12 months after cervical spine surgery. Siccoli et al. employed eight ML models to predict the ODI and NRS scores for BP and LP [31]. The 6-week postoperative AUC values were as follows: ODI 0.75, NRS LP 0.79, and NRS-BP 0.92 (boosted trees model). At 12 months postoperatively, the AUC values were ODI 0.68, NRS-LP 0.72, and NRS-BP 0.79. Ames et al. used seven ML models to predict individual SRS-22R questions, achieving AUROC values for individual SRS-22R questions as high as 86.9% (extreme gradient boosting tree) [22]. Staartjes et al. used EN regularization to predict the ODI and COMI, achieving an AUC of 0.67 [32]. The model yielded an AUC of 0.72 for BP and 0.64 for LP. Khan et al. utilized a polynomial SVM model to predict an AUC of 0.834 [18]. Khor et al. applied three binary regression models to predict outcomes, achieving the following AUC values: ODI 0.66, BP 0.79, and LP 0.69 [24]. Hoffman et al. reported a mean absolute accuracy (MAA) of 0.0283 with the use of support vector regression (SVR) [19].

## 4. Discussion

Predicting clinically relevant outcomes after spine surgery has been increasingly performed with patient-reported outcomes [35]. These questionnaires evaluate subjective and objective measures that aid surgeons in measuring a patient’s quality of life before and after surgical intervention, ultimately allowing for a better understanding of the physical and psychological burden of spinal pathology. By identifying subtle patterns in pathology, patient characteristics, and populations, ML has the potential to predict PROMs after spine surgery. There has been a significant volume of studies describing PROMs, yet the clinical relevance has yet to be determined due to the significant degree of heterogeneity [35]. To improve the consistency and completeness of prediction model studies, the Transparent Reporting of a multivariable prediction model for Individual Prognosis Or Diagnosis (TRIPOD) was devised. This TRIPOD criteria serves as a set of evidence-based guidelines designed to improve the consistency and completeness of prediction model reporting [36]. Only eleven studies [15,17,20,22,23,26,27,31,32,34] in the review claimed to adhere to the TRIPOD criteria.

PROMs often assess pain, functional status, and other relevant factors. Consistent with past literature reviews [6], the mJOA, ODI, and SRS-22 were the most frequently predicted PROMs for cervical, lumbar, and spinal deformity pathologies, respectively. This fact highlights the emphasis placed on a patient’s physical function. For assessment of pain, tools like the VAS and NRS are commonly used to measure back and leg pain. Although both measure pain, some studies have found the VAS assessment to be more useful. For example, Bielewicz et al. found that VAS scores decreased to a greater degree than NRS scores three months after surgery [37], attributing the poor reproducibility of the NRS to its less detailed incremental changes [37].

The features used to predict PROMs included demographics, surgery characteristics, preoperative PROMs, spinal pathology characteristics, mental health evaluations, employment details, social history, ASA classification, comorbidities, imaging findings, fine motor function, hospital characteristics, and surgeon characteristics. Several physical exam findings have been identified as predictors of functional improvement after surgery [16]. For example, upper motor neuron signs have been associated with a decreased likelihood of recovery after lumbar spine surgery [18]. In addition to objective clinical findings, Finkelstein et al. found that cognitive factors accounted for 40% of the variance in PROMs after spine surgery [30]. This finding is consistent with a randomized control trial reporting that lumbar spine surgery patients who participated in cognitive behavioral-based physical therapy had greater improvements in pain and disability compared to those who received physical therapy-related educational training [38]. Preoperative opioid use has been identified as another factor that affects patient-reported outcomes after spine surgery. Given that unmanageable pain is often a primary reason for surgical intervention [39], this variable should be further investigated for its role in predicting PROMs.

In this report, the AUC was the most frequently reported performance metric. The AUC can be thought of as the overall performance of an ML model with values ranging from 0 to 1. Values closer to 1 indicate better performance [40]. The study reporting the highest AUC for cervical spine pathology was that by Khan et al., who used a polynomial SVM to predict mJOA with an AUC of 0.834 [18]. Siccoli et al. applied boosted trees to predict NRS-BP after lumbar spine surgery, achieving an AUC of 0.92 [31]. For adult spinal deformity, Ames et al. used extreme gradient boosting trees to predict individual SRS-22R questions, with AUC values as high as 0.869 [22]. Despite successful ML model performance, the clinical applicability of these models is limited due to the complexity of shared decision making between a patient and the provider. A review by Christodoulou et al. evaluated 71 studies investigating clinical prediction models and found no evidence of superior ML performance over LR [41].

A primary limitation of this review was the exclusion of articles not containing the term “machine learning” in the abstract or title. This may have excluded studies that employed ML to predict PROMs but did not explicitly mention “ML” in their terminology. As a result, this introduced a potential selection bias and reduced the overall comprehensiveness of the review. Another contributing limitation was the minimal volume of high-quality evidence. Due to the negligible amount of evidence and large degree of heterogeneity amongst studies, a comprehensive systematic review or meta-analysis was unable to be performed.

## 5. Conclusions

PROMs continue to be a valuable tool for assessing the impact of spine pathology on physical and mental health, but surgeon expertise remains pivotal when counseling patients. Providers should be aware of the evolving application of these technologies in both clinical and academic pursuits. Although ML models have the potential to accurately predict PROMs, their clinical applicability is severely limited by the variation in ML models, spinal pathology, input variables, and output variables across studies. Surgeons and researchers should collaborate to establish standardized outcome measures and evaluation metrics. This joint effort would harness the predictive potential of ML to predict postoperative PROMs.

## Figures and Tables

**Figure 1 bioengineering-12-00125-f001:**
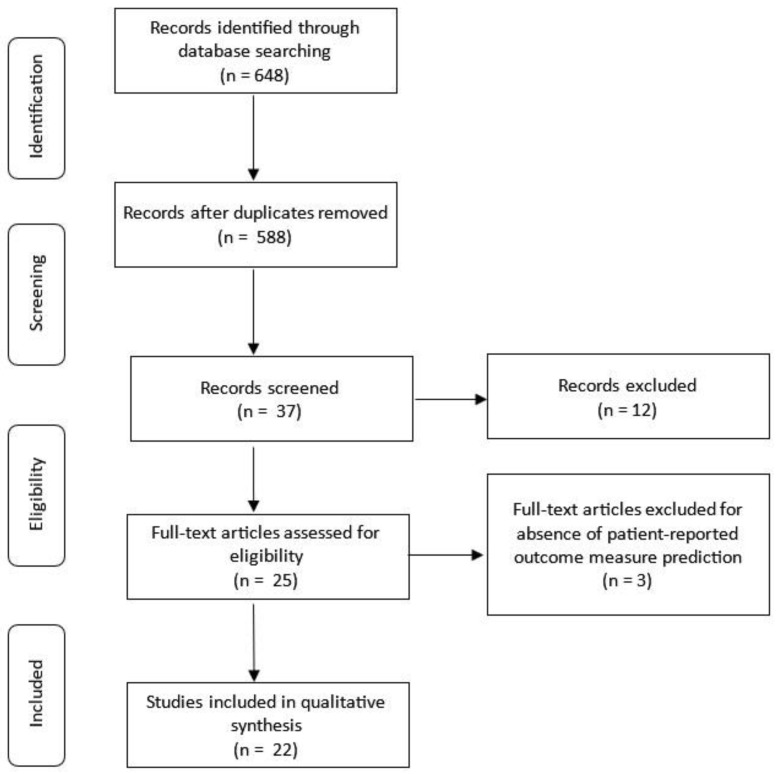
Literature search strategy.

**Table 1 bioengineering-12-00125-t001:** Study characteristics.

Article	Pathology	# of Pts	Predicted PROMs	ML Models	Input Features	PPT (Months)	Results (AUC) *
Liew et al. [17]	Cervical	193	NDI, EQ5D, NP	Stepwise regression, LASSO, boosting, MARS	Demographics, PE, PROMs	12	Not reported
Park et al. [13]	CSM	535	VAS-NP	LR, SVM, DT, RF, extra trees, Gaussian naïve Bayes, KNN, multilayer perceptron, EGBT	Demographics, Sx chars, PROMs, spinal pathology, PE	3; 24	VAS-NP 0.773–0.762
Zhang et al. [15]	CSM	50	SF-36, PCS	SVM	Demographics, PE, PROs, imaging chars	6	SF-36 PCS 86.4, MCS 89.8
Merali et al. [14]	DCM	757	SF-6D, mJOA	RF, SVM, LR, DT, ANN models	Demographics, spinal pathology, Sx chars, comorbidities, PROMs, PE	3–24	SF-6D and mJOA0.83–0.87
Khan et al. [16]	DCM	173	SF-36 MCS, SF-36 PCS	Classification trees, SVM, partial least squares, generalized boosted models, generalized additive models, MARS, RF, LR	Demographics, PE, comorbidities, Sx Hx, spinal pathology, mJOA	12	MCS 0.77, PCS 0.78
Hoffman et al. [19]	DCM	20	ODI	MLR, SVR	Demographics, spinal pathology, Sx chars, comorbidities, PROs, PE, fine motor function	6, 12, and 24	MAA of 0.0283 with SVR
Khan et al. [18]	DCM	702	mJOA	Boosted LR, SVM, naïve Bayes, generalized boosted machines, partial least squares, LR	Demographics, Sx chars, spinal pathology, PE	12	mJOA 0.834
Grob et al. [20]	Thoraco- lumbar	1115	ODI, BP (NRS), LP (NRS)	FUSE-ML, EN regularization	MRI, PROMs, demographics, ASA, PMHx, Sx Hx	12	ODI 0.70, BP 0.72, LP 0.70
Gupta et al. [21]	AIS	6076	SRS-Pain, SRS-Self-Image	LR, gradient boosting, EGBT	PROMs, demographics, spinal pathology, Sx chars	6; 12; 24	MAE 0.47–0.55
Pedersen et al. [25]	LDH	1968	ODI, VAS	DL, DT, RF, BT, SVM, LR, MARS	Demographics, PROs, employment details, comorbidities, self-reported expectations to return to work	24	EQ-5D 0.82, ODI 0.75, VAS LP 0.73, VAS BP 0.81, return to work 0.84
Ve et al. [23]	LDH	422	LP (NRS), BP (NRS), ODI	DL, LR	Demographics, ASA, PROMs, Sx chars, Sx Hx, spinal pathology, social Hx	12	BP 0.90, LP 0.87, ODI 0.84
Ames et al. [22]	ASD	561	Individual SRS-22R questions	EN, gradient boosting machines, EGBT, extreme gradient boosting linear, RF, EN regularized generalized linear models	Demographics, comorbidities, Sx chars, imaging chars, hospital chars, surgeon chars	12	SRS-22R questions 0.869 with EGBT
Karhade et al. [28]	LS	906	PROMIS-PF	Stochastic gradient boosting, RF, SVM, NN, EN penalized LR	Demographics, ASA, spinal pathology, Sx chars, PROMs, Rx opioids, geographic information	12	PROMIS-PF 0.75
Yagi et al. [29]	LS	848	VAS BP, VAS LP, JOABPEQ	Generalized LR, generalized linear mixed, LR, SVM, single-layer ANN, random trees, linear-AS, tree-AS, EGBT, chi-squared automatic interaction detection classification, regression tree	Demographics, Sx chars, PROMs	10	MAE 9.3−16.5
Siccoli et al. [31]	LS	635	BP (NRS), LP (NRS), ODI	RF, EGBT, BGLM, BT, KNN, simple BGLM, ANN with a single hidden layer	Clinical data, imaging chars, PROMs, demographics, ASA, Sx Hx, spinal pathology	6 weeks; 12 months	NRS-BP 0.79, 0.92
Khor et al. [24]	LS	1965	BP (NRS), LP (NRS), ODI	Binary LR	Demographics, clinical chars, ASA, Sx Hx, PROMs, comorbidities, Sx chars, Rx opioids, hospital chars	12	ODI 0.66, BP 0.79, and LP 0.69
Berjano et al. [26]	Lumbar	1243	ODI, SF-36, PCS, COMI Back	RF	Demographics, comorbidities, spinal pathology, PROMs, past Sx Hx	6	ODI 0.808
Finkelstein et al. [30]	Lumbar	122	NRS	LASSO regression	Clinical and demographic variables, PROMs, patient expectations and cognitive appraisal processes	10	NRS of 0.12 MBR2
Staartjes et al. [32]	Lumbar	1115	ODI, COMI, NRS	EN regularization	Demographics, Rx opioids, Sx Hx, Sx chars, PROMs	12	ODI and COMI 0.67
Halicka et al. [27]	Lumbar	4307	COMI-BP, COMI-LP	RF, LR	Demographics, Sx chars, hospitalization chars	3–24	COMI 0.63, BP 0.72, LP 0.68
Rigoard et al. [33]	Lumbar	200	PGIC	DRFA, PCA	ODI, EQ-5D, HADS, NRS	12	PGIC 0.853
Muller et al. [34]	Cervical and lumbar	10,002	COMI	LASSO, ridge regression	Demographics, Sx chars, surgeon chars, PROMs, psychological assessment	12	MAE back patients 2.1, neck patients 1.8

* = unless otherwise specified; # = number; AIS = adolescent idiopathic scoliosis; ANN = artificial neural network; ASA = American Society of Anesthesiologist; ASD = adult spinal deformity; AUROC = area under the receiver operating characteristic curve; BGLM = Bayesian generalized linear model; BP = back pain; BT = boosted tree; CSM = cervical spondylotic myelopathy; COMI = core outcome measure index; DRFA = dimensionality reduction factor analysis; DT = decision tree; EGBT = extreme gradient boosting tree; EN = elastic net; EQ-5D = EuroQol–5 dimensions; HADS = the hospital anxiety and depression scale; Hx = history; JOABPEQ = Japanese Orthopedic Association back pain evaluation questionnaire; KNN = k-nearest neighbors; LASSO = least absolute shrinkage and selection operator; LDH = lumbar disk herniation; LP = leg pain; LR = logistic regression; MAA = mean absolute accuracy; MAE = mean absolute error; MARS = multivariate adaptive regression spline; MBR2 = mean bootstrapped R2; MCS = Mental Component Summary; MCRI = multidimensional clinical response index; mJOA = modified Japanese Orthopaedic Association; MLR = multivariate linear regression; NDI = Neck Disability Index; NP = neck pain; NRS = numeric rating scale; PE = physical exam; PCA = principal component analysis; PCS = Physical Component Summary; PGIC = Patient Global Impression of Change; PMHx = past medical history; Pts = patients; PPT = postoperative prediction timeline; PROMIS-PF = Patient-Reported Outcomes Measurement Information System-Physical Function; RF = randomforest; Rx = prescription; SF-6D = short form-6 dimensions; SF-36 = short form-36 health survey; SRS = Scoliosis Research Society; SVR = support vector regression; SVM = support vector machine; Sx = surgical; VAS = Visual Analog Scale.

**Table 2 bioengineering-12-00125-t002:** Description of common patient-reported outcome measures.

Domain	PROM	Description
Multiple Outcomes	MCRI	Modified Clinical Response Index (MCRI) evaluates pain, functional capacity, quality of life, and outcomes in spinal surgery patients with Persistent Spinal Pain syndrome
	NASS	North American Spine Society (NASS) assesses outcomes and pain related to lumbar spine disease
	EQ-5D	EuroQol-5 Dimensions (EQ-5D) measures health status across five dimensions: mobility, self-care, usual activities, pain/discomfort, anxiety/depression
	COMI	Core Outcome Measures Index (COMI) measures the impact of back and leg pain, assessing pain, function, and quality of life
	SRS	Scoliosis Research Society (SRS) assesses function, pain, self-image, mental health, and satisfaction
Physical Function	NDI	Neck Disability Index (NDI) evaluates disability related to neck pain and its impact on daily activities
	JOA	Japanese Orthopaedic Association Score (JOA) assesses neurological function in patients with cervical myelopathy
	mJOA	Modified JOA (mJOA) evaluates functional impairment in cervical spine conditions
	ODI	Oswestry Disability Index (ODI) assesses disability due to lower back pain
	PROMIS- PF	Patient-Reported Outcomes Measurement Information System (PROMIS)-Physical Function (PF) assesses physical function and the ability to perform physical activities
	PCS	Physical Component Summary (PCS) is a subscore from SF36 measuring physical health
	DASH	Disabilities of the Arm, Shoulder, and Hand (DASH) measures upper-extremity function, pain, and work and social activity participation
Mental Health	MCS	Mental Component Summary (MCS) assesses psychological well-being
	MDI	Mental Disability Index (MDI) measures mental health-related disability
	PGIC	Patient Global Impression of Change (PGIC) measures a patient’s overall perception of improvement or change in condition
Quality of Life	SF-36	Short Form-36 Health Survey (SF-36) assesses overall health-related quality of life across multiple domains (physical, mental, and social)
	SF-6D	Short Form-6 Dimensions (SF-6D) is a condensed version of SF36 that measures a single index for health-related quality of life
Pain	VAS	Visual Analog Scale (VAS) measures intensity of pain using a 0–10 visual scale
	NRS	Numeric rating scale (NRS) quantifies pain on a 0–10 scale
Social	JOABPEQ	Japanese Orthopaedic Association Back Pain Evaluation Questionnaire (JOABPEQ) evaluates the impact of back pain on physical and social functioning

## Data Availability

No new data were created or analyzed in this study. Data sharing is not applicable to this article.

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
