# Peer review of "Scoping Review of Machine Learning and Patient-Reported Outcomes in Spine Surgery"

_bioengineering, 2025, doi:10.3390/bioengineering12020125_

Round 1

Reviewer 1 Report

Comments and Suggestions for Authors

The authors present a review of applications of machine learning and patient reported outcomes in spinal surgery. The topic is certainly interesting and relevant, as there have been in the recent years a large explosion in the interest towards these applications. However, at the same time, there are some aspects that should be clarified about the survey design, and the current usefulness of the review itself seems limited, as the findings and related discussion is very basic. Some more detailed comments:

- The criteria for inclusion and exclusion of article should be better specified since some relevant articles seem to have been excluded, e.g.

Campagner, A., et al. (2024). Second opinion machine learning for fast-track pathway assignment in hip and knee replacement surgery: the use of patient-reported outcome measures. BMC Medical Informatics and Decision Making, 24(Suppl 4), 203.

which seems to agree to the inclusion criteria (prediction of post-surgery PROMs, using SF-12 PROM).

- Also, why different queries were given to different search engines? Why other relevant engines (e.g., Scopus) were not used?

- The insights that could be drawn from the results seems quite limited, as the authors mostly simply report results across studies without specific comments about studies' validity or reproducibility. I believe that the authors could make their article more relevant and useful for readers by providing a more insightful analysis: for example, they could provide an analysis of reporting, validation and reproducibility in the literature, guided by some checklist or guidelines such as:

Vasey, B., et al. (2022). Reporting guideline for the early stage clinical evaluation of decision support systems driven by artificial intelligence: DECIDE-AI. bmj, 377.

Cabitza, F., et al. (2021). The need to separate the wheat from the chaff in medical informatics: Introducing a comprehensive checklist for the (self)-assessment of medical AI studies. International Journal of Medical Informatics, 153, 104510.

Author Response

Author 1 comments:

Comment 1: The criteria for inclusion and exclusion of article should be better specified since some relevant articles seem to have been excluded, e.g.

Campagner, A., et al. (2024). Second opinion machine learning for fast-track pathway assignment in hip and knee replacement surgery: the use of patient-reported outcome measures. BMC Medical Informatics and Decision Making, 24(Suppl 4), 203. Which seems to agree to the inclusion criteria (prediction of post-surgery PROMs, using SF-12 PROM.

Response 1:Thank you for your feedback. I will emphasize the inclusion criteria of patients undergoing spine surgery. The text reads: “The inclusion criteria consisted of studies that utilized ML tools to predict postoperative PROMs for patients who underwent spine surgery.”. Now that I have emphasized the inclusion criteria of spine surgery, this study would not fall under the inclusion criteria.

Comment 2: Also, why different queries were given to different search engines? Why other relevant engines (e.g., Scopus) were not used? 
Response 2: Thank you for pointing this out. We found the use of three seach engines (Web of Science, PubMed, and EMBASE) to be sufficient. As a scoping review, we believe that three major search engines were sufficient. The combination of MESH terms and keywords were not identical because search engines do not have ubiquitous search language. 

Comment 3: The insights that could be drawn from the results seems quite limited, as the authors mostly simply report results across studies without specific comments about studies' validity or reproducibility. I believe that the authors could make their article more relevant and useful for readers by providing a more insightful analysis: for example, they could provide an analysis of reporting, validation and reproducibility in the literature, guided by some checklist or guidelines such as:

Vasey, B., et al. (2022). Reporting guideline for the early stage clinical evaluation of decision support systems driven by artificial intelligence: DECIDE-AI. bmj, 377. 

Cabitza, F., et al. (2021). The need to separate the wheat from the chaff in medical informatics: Introducing a comprehensive checklist for the (self)-assessment of medical AI studies. International Journal of Medical Informatics, 153, 104510.

Response 3: Thank you for your suggestions and for recommending specific AI literature assessment sources. Our goal for this manuscript is to report the results across studies. As far as validity or reproducibility goes, the variability in the type of surgery, predicted PROMs input and output variables varies to such a large degree, that comparing the studies would be unfeasible.  Our discussion points out that although ML has been applied to predict PROMs, there is extensive heterogeneity and that external validity and reproducibility of these PROMs are questionable.

Reviewer 2 Report

Comments and Suggestions for Authors

Thank you for the opportunity to review this manuscript.

  1. Please provide a clear and concise objective for the study at the end of the introduction.
  2. The manuscript states, “A systematic review of the literature per Preferred Reporting Items for Systematic Reviews and Meta-Analyses (PRISMA) guidelines [8] was carried out in Web of Science, PubMed, and EMBASE on October 8, 2024.” Could the authors clarify whether this study is a systematic review or a scoping review? These study types have distinct purposes and methodologies, and it is important to use consistent terminology throughout the manuscript.
  3. The methods section states, “A combination of MeSH terms and keywords related to patient-reported outcomes and spine surgery were used.” Unfortunately, Appendix A is not accessible. The exact search strategy, including all MeSH terms and keywords used, should be presented within the methods section for transparency and reproducibility.
  4. The manuscript mentions “spine surgery and utilized ML tools to predict postoperative PROMs.” Could the authors clarify which machine learning methods were included? There are numerous machine learning approaches, and some studies may use such methods without explicitly labeling them as “machine learning.” Please specify the exact methods included in the analysis.
  5. Regarding data extraction, who was responsible for extracting the data? Was the process automated, human-based, or a combination of both? Additionally, what specific information was extracted from the included studies?
  6. Did the authors assess the quality of the included studies? If so, please provide details of the criteria or tools used for the assessment.
  7. Table 1 appears incomplete, as only the left portion of the first page is visible. Please ensure that the entire table is accessible and clearly formatted.
  8. Did the authors extract performance metrics such as sensitivity, specificity, positive predictive value (PPV), negative predictive value (NPV), and area under the curve (AUC) from the included studies? If not, could the authors justify why these critical metrics were not considered?
  9. Why was a meta-analysis not performed despite the availability of an acceptable number of studies in each group? Please elaborate on the rationale.
  10. Finally, please include a flowchart detailing the study selection process, including the number of studies included and excluded at each stage. This is essential for transparency and adheres to PRISMA guidelines.

Author Response

Author 2 comments:

Comment 1: Please provide a clear and concise objective for the study at the end of the introduction. 

Response 1: Thank you for pointing this out. We have not added the following to the text: “The purpose of this manuscript is to describe the breadth of application of machine learning to predict PROMs after spine surgery.”

Comment 2: The manuscript states, “A systematic review of the literature per Preferred Reporting Items for Systematic Reviews and Meta-Analyses (PRISMA) guidelines [8] was carried out in Web of Science, PubMed, and EMBASE on October 8, 2024.” Could the authors clarify whether this study is a systematic review or a scoping review? These study types have distinct purposes and methodologies, and it is important to use consistent terminology throughout the manuscript.

Response 2: My apologies for this error. This text now reflects that this scoping review was carried out as per the PRISMA-Scoping review guidelines.

Comment 3: The methods section states, “A combination of MeSH terms and keywords related to patient-reported outcomes and spine surgery were used.” Unfortunately, Appendix A is not accessible. The exact search strategy, including all MeSH terms and keywords used, should be presented within the methods section for transparency and reproducibility. 

Response 3: Thank you for pointing this out. I have included the search criteria for each data base in the methods section:
The search criteria for Pubmed: ("Machine Learning"[MeSH] OR "Artificial Intelligence") AND ("Patient Reported Outcome Measures"[MeSH] OR "Patient-reported outcomes"). The search criteria for Web of Science: ("Machine Learning" OR "Artificial Intelligence") AND ("Patient Reported Outcome Measures" OR "Patient-reported outcomes" OR "PROMs" OR "Quality of Life" OR "Health Outcomes") AND ("Spine" OR "Spinal Surgery"). The search criteria for EMBASE: ('machine learning'/exp OR 'machine learning' OR 'artificial intelligence'/exp OR 'artificial intelligence') AND ('patient reported outcome'/exp OR 'patient reported outcome' OR 'quality of life'/exp OR 'quality of life' OR 'patient-reported outcomes' OR 'proms' OR 'health outcomes'/exp OR 'health outcomes') AND ('spine'/exp OR 'spine' OR 'spinal surgery'/exp OR 'spinal surgery' OR 'spine surgery'/exp OR 'spine surgery') AND [english]/lim.

Comment 4: The manuscript mentions “spine surgery and utilized ML tools to predict postoperative PROMs.” Could the authors clarify which machine learning methods were included? There are numerous machine learning approaches, and some studies may use such methods without explicitly labeling them as “machine learning.” 

Response 4: I agree with the observation that manuscripts may have employed machine learning without labeleing them as such. Unfortunately, our review only included articles whose title and/or abstract specified the use of “machine learning”. As far as which specific ML models were included, all ML models mentioned in the included articles were discussed. 

Comment 5: Please specify the exact methods included in the analysis. 
Response 5: As a scoping review, the summary of evidence was summarized in the discussion and included a breakdown of the most commonly reported PROMs as well as the most common features used to predict postoperative PROMs.

Comment 6: Regarding data extraction, who was responsible for extracting the data?  Was the process automated, human-based, or a combination of both? Additionally, what specific information was extracted from the included studies? 

Response 6: I apologize for not specifying which research member assessed the articles for eligibility. The manuscript now reads: “One researcher (C.Q) assessed manuscripts for eligibility under the supervision of (D.K.). In cases requiring futher clarification, the senior author (S.H.) was consulted for their expertise.”

Comment 7: Did the authors assess the quality of the included studies? If so, please provide details of the criteria or tools used for the assessment. 

Reponse 7: As a scoping review, the quality of the included studie was not assessed. 

Comment 8: Table 1 appears incomplete, as only the left portion of the first page is visible. Please ensure that the entire table is accessible and clearly formatted. 

Response 8: Thank you for brining this to our attention. Our MS word document and the submitted pdf has the entirety of table 1 visible. I apologize for this inconvenience and we will reach out to the MDPI team about this. 

Comment 9: Did the authors extract performance metrics such as sensitivity, specificity, positive predictive value (PPV), negative predictive value (NPV), and area under the curve (AUC) from the included studies? If not, could the authors justify why these critical metrics were not considered? 

Response 9: Under section 3.3 Key Results, the performance metrics of each study is expanded upon. Additionally, the final column in Table 1 details the reported results of the ML models.

Comment 10: Why was a meta-analysis not performed despite the availability of an acceptable number of studies in each group? Please elaborate on the rationale. 

Response 10: Although 22 articles were included in the study, the predictied PROMs for each pathology differed greatly. For example, lumbar disc herniation and degenerative cervical myelopathy were the most studied pathology yet each was described in merely 4 studies. Due to this, the research team did not believe a meta-analysis to be feasible. 

Comment 11: Finally, please include a flowchart detailing the study selection process, including the number of studies included and excluded at each stage. This is essential for transparency and adheres to PRISMA guidelines. 

Response 11: Thank you for bringing this to our attention. As with the formatting issue with Table 1, Figure 1 (see below) was created to detail the study selection process. We will reach out to the MDPI team to ascertain why you were unable to visualize Figure 1. 

Round 2

Reviewer 2 Report

Comments and Suggestions for Authors

Thank you for responding to my queries. Please ensure that these final comments are reflected in your manuscript.

 “Response 4: I agree with the observation that manuscripts may have employed machine learning without labeleing them as such. Unfortunately, our review only included articles whose title and/or abstract specified the use of “machine learning”. As far as which specific ML models were included, all ML models mentioned in the included articles were discussed. “ Please add this point to the limitations section of your study.

“The manuscript now reads: “One researcher (C.Q) assessed manuscripts for eligibility under the ‎supervision of (D.K.). In cases requiring futher clarification, the senior author (S.H.) was consulted for ‎their expertise.”‎:

1-Using only one researcher for eligibility screening introduces a risk of bias or errors.

2- Most scoping review methodologies (e.g., PRISMA-ScR) recommend dual independent screening of manuscripts to enhance reliability and reproducibility.

3- The manuscript should explicitly clarify how disagreements or uncertainties were resolved.

4- These points should be included in the limitations section of the study.

Author Response

Comments 1: Using only one researcher for eligibility screening introduces a risk of bias or errors.
Response 1: Thank you for taking the time to provide your thoughtful insight. We appreciate the reviewer's suggestion; the revised text reads as follows "A primary limitation in this review was the exclusion of articles not containing the term "machine learning" in the abstract or title. This may have omitted studies that used ML to predict PROMs, but the term ML was not explicitly stated. Thus, introducing selection bias and limiting the comprehensiveness of the review." (lines 241-244)

Comments 2: Most scoping review methodologies (e.g., PRISMA-ScR) recommend dual independent screening of manuscripts to enhance reliability and reproducibility.
Response 2: Thank you for reminding us of the importance of dual independent screening. We acknowledge the eligibility screening being conducted by a single researcher as a limitation in our methodology. Please see lines 243-244 “Thus, introducing selection bias and limiting the comprehensiveness of the review.” 

Comments 3: The manuscript should explicitly clarify how disagreements or uncertainties were resolved.
Response 3: Thank you for pointing this out. The revision located at lines 70-72 now reads "In cases of disagreements or uncertainties requiring further clarification, the senior author (S.H.) was consulted and a consensus was reached during research team discussions."

Comments 4: These points should be included in the limitations section of the study.
Response 4: The changes above reflect the latest version of the manuscript. We trust that they adequately address your comments. Thank you again for the opportunity to improve our manuscript.